

# Sensitivity of a collisional single-atom spin probe

Jens Nettersheim[1], Quentin Bouton[1,2], Daniel Adam[1] and Artur Widera[1*]

**1** Department of Physics and Research Center OPTIMAS,
Technische Universität Kaiserslautern, Germany
**2** Laboratoire de Physique des Lasers, CNRS, UMR 7538,
Université Sorbonne Paris Nord, F-93430 Villetaneuse, France

* widera@physik.uni-kl.de

## Abstract

We study the sensitivity of a collisional single-atom probe for ultracold gases. Inelastic spin-exchange collisions map information about the gas temperature $T$ or external magnetic field $B$ onto the quantum spin-population of single-atom probes, and previous work showed enhanced sensitivity for short-time nonequilibrium spin dynamics [1]. Here, we numerically investigate the steady-state sensitivity of such single-atom probes to various observables. We find that the probe shows distinct sensitivity maxima in the $(B, T)$ parameter diagram, although the underlying spin-exchange rates scale monotonically with temperature and magnetic field. In parameter space, the probe generally has the largest sensitivity when sensing the energy ratio between thermal energy and Zeeman energy in an externally applied magnetic field, while the sensitivity to the absolute energy, i.e., the sum of kinetic and Zeeman energy, is low. We identify the parameters yielding sensitivity maxima for a given absolute energy, which we can relate to a direct comparison of the thermal Maxwell-Boltzmann distribution with the Zeeman-energy splitting. We compare our equilibrium results to nonequilibrium experimental results from a single-atom quantum probe, showing that the sensitivity maxima in parameter space qualitatively prevail also in the nonequilibrium dynamics, while a quantitative difference remains. Our work thereby offers a microscopic explanation for the properties and performance of this single-atom quantum probe, connecting thermodynamic properties to microscopic interaction mechanisms. Our results pave the way for optimization of quantum-probe applications in $(B, T)$ parameter space beyond the previously shown boost by nonequilibrium dynamics.

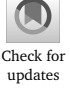

# 1  Introduction

Local probing of quantum systems at ultracold temperature is a prominent challenge in quantum technology and computing application [2,3]. In this context, recent theoretical proposals have suggested superior performances of novel types of quantum probes [4–10]. Concomitantly, experimental advances have led to the immersion of small systems as probes, such as single ions [11], single atoms [12], small confined BEC [13], or Fermi sea [14] inside ultracold gases. These probes allow storing information about the many-body system like the temperature [13,15], the surrounding magnetic field [1], or the density [16] in quantum observables with an unprecedented precision. Furthermore, they show highly desirable properties compared to their classical counter parts, including a minimal perturbation of the many-body system or a precision below the standard quantum limit [17]. In particular, one of the figures of merit in quantifying the performance of a quantum sensor is its sensitivity to the observables probed, where a quantum sensitivity enhancement has been demonstrated for collisional spin-exchange probes out of equilibrium [1]. This raises the question if the performance and specifically the sensitivity of such quantum probes can be understood from the microscopic interaction mechanisms determining individual atomic collisions. Such understanding could open the door to further optimization of the probing process.

In this work, we consider a single neutral Cs impurity atom as quantum probe immersed in an ultracold Rb bath, see Fig. 1(a), originally introduced in Ref. [1]. Impurity and bath can exchange single quanta of angular momentum via inelastic spin-exchange (SE) collisions. Owing to angular momentum conservation, these SE processes can be divided into exoergic collisions (promoting Cs atoms to energetically higher lying magnetic sub-states) and endoergic processes (promoting Cs atoms to energetically lower lying magnetic sub-states) [1], see Fig. 1(b). Starting from an initial state of the probe, both processes change the spin state of the probe. Furthermore, due to energy conservation, the endoergic rate specifically strongly depends on the temperature of the gas and the externally applied magnetic field. Endo- and exothermal SE collisions thereby provide a tool for sensing the bath temperature or an external magnetic field by mapping bath information onto the internal quantum states. Importantly, they yield enhanced sensitivity based on nonequilibrium spin dynamics [1].

Here, we will consider the steady-state performance of such probes to obtain an intuitive understanding of how the microscopic collision mechanisms are related to the quantum probe

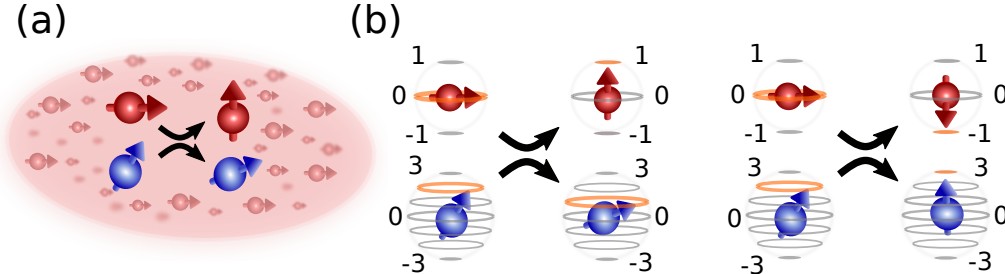

Figure 1: **Single Cs atom immersion in an ultracold Rb bath and microscopic interaction mechanisms.** (a) Impurity atom (blue) inside the ultracold Rb bath (red). Exemplarily one SE collision is shown. Internal Zeeman level structures with three (seven) states for Rb (Cs) are presented as Bloch spheres in (b). Current atom states are indicated by the spin tilt and highlighted in orange on the spheres to illustrate both spin-exchange processes exoergic in the left and endoergic in the right part.

sensitivity. The steady-state regime is independent of the initial state and the evolution time so that the parameter space we consider in the following is reduced and not affected by the nonequilibrium enhancement of the sensitivity. We find that the sensitivity exhibits distinct maxima in $(B, T)$ parameter space, which we relate to the microscopic competition of thermal and Zeeman energy.

The quantum spin dynamics of the impurity-probe system and its steady state are determined by the Zeeman energy $E_Z$ and thermal energy $E_{th}$. In particular, the rate of endothermal spin-exchange collisions strongly depends on the ratio between thermal and magnetic-field energy $E_{ratio} = E_{th}/E_Z$ for a given total energy $E_{tot} = E_Z + E_{th}$. In fact, sensing one of these four quantities corresponds to one of the four sensing applications of calorimetry ($E_{tot}$), thermometry (i.e., temperature $T = E_{th}/k_B$ with Boltzmann constant $k_B$), magnetometry (i.e., magnetic field $B = E_Z/\mu_B$, with the Bohr magneton $\mu_B$), and the ratio of energy contributions $E_{th}/E_Z$. Total energy and energy ratio, on the one hand, and thermal energy and Zeeman energy, on the other hand, form pairs of independent parameters and span the parameter range of sensing as orthogonal axes, as shown in Fig. 2.

In order to characterize the performance of the probe, we focus on the sensitivity by calculating the Fisher information that is often used for parameter estimation [18]. We use a detailed rate model simulating the quantum spin dynamics for a range of parameters. Thereby, we compute the sensitivity of our single-atom spin sensor to thermal energy $E_{th}$ or Zeeman energy $E_Z$, total energy $E_{tot}$, or energy ratio $E_{ratio}$. We find that the probe is best suited, i.e., has increased sensitivity, to sense the ratio between thermal and magnetic-field energy $E_{ratio}$, while it has slightly smaller sensitivity to temperature $T$ and magnetic field $B$, and it is almost insensitive as calorimeter sensing the total energy $E_{tot}$ of the system. We identify the parameters exhibiting maximum sensitivity and find that they are related to the functional form of the probability distribution for endothermal collisions, which originates from the competition of thermal and Zeeman energies in the process of endoergic SE collision. Finally, we compare our theory to experimental data and find similar qualitative behaviors. This link between microscopic interaction processes and the macroscopic performance of the sensor offers a way to predict optimal strategies or parameter optimization for probing ultracold gases.

## 2 Microscopic probing mechanism

The central mechanism mapping information about the bath temperature or magnetic field onto the probe relies on inelastic SE collisions between the $^{133}$Cs impurity in $|F_{Cs} = 3, m_{F,Cs} = 2\rangle$ and $^{87}$Rb bath atoms in the $|F_{Rb} = 1, m_{F,Rb} = 0\rangle$ state. Here, $F$ and $m_F$ denote the total angular momentum and its projection on the quantization axis, respectively, with $m_{F,Cs} \in [3, 2, ..., -3]$ constituting the total Cs spin space available for the quantum probe. SE processes modify the spin state $|F_{Cs}, m_{F,Cs}\rangle$ and $|F_{Rb}, m_{F,Rb}\rangle$. An endoergic SE collision transfers a single quantum of angular momentum from a bath to the probe atom ($|F_{Cs}, m_{F,Cs}\rangle \rightarrow |F_{Cs}, m_{F,Cs} + 1\rangle$ and $|F_{Rb}, m_{F,Rb}\rangle \rightarrow |F_{Rb}, m_{F,Rb} - 1\rangle$). The ensuing spin dynamics and the steady state are fully inscribed by the SE rates $\Gamma^{m_F}$, which are dominated by the competition between the collision energy $E_C = \mu v_{rel}^2/2$ in each collisional event, and the Zeeman energy given by $E_Z = \mu_B g_F B$ for a small external magnetic field considered in this work. Here $\mu$ is the reduced mass, $v_{rel}$ the relative velocity of the colliding atoms, and $g$ the Landé factor. Since the Landé factors of the species used differ by a factor of two, $g_{F,Rb} = 2g_{F,Cs}$, the Zeeman-energy quantum taken by one collision partner does not match the Zeeman-energy provided by the other, as illustrated for endoergic events in the insets (a)-(d) of Fig. 2. As a consequence, in an exothermal

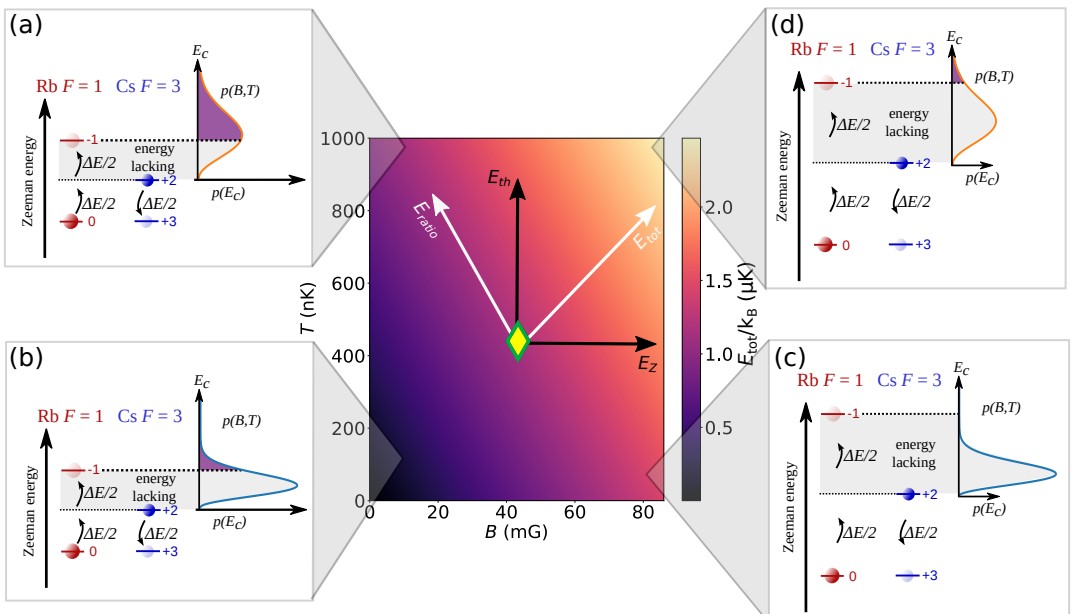

Figure 2: **Parameter space of single-atom spin probing.** The parameter space is spanned by the pairs $B, T$, and $E_{tot}, E_{ratio}$ of independent variables to be sensed. They can be visualized as a coordinate system, where for a given point to be sensed, one parameter is kept fix, while the independent parameter (orthogonal axis) is varied. The total energy $E_{tot}$ is minimal for low magnetic fields and temperatures (lower left corner) and increases when the magnetic field and/or temperature increase. The four sensitivity directions are depicted in the coordinate systems. The diamond marks the reference point at which the sensitivity is evaluated in the following along the four axes of parameter space. Panels (a)-(d) show the energetic competition between thermal and Zeeman energies for endothermal SE processes at selected points in parameter space, and the corresponding contribution to the spin dynamics.

(endothermal) collision, the energy difference

$$\Delta E/2 = \mu_{\mathrm{B}} |g_{\mathrm{F,Cs}}| B \tag{1}$$

increases (decreases) the kinetic energy of the colliding atoms.

Concretely, endothermal collisions can only occur if the missing energy fraction $\Delta E/2$ can be provided by the thermal collisional energy and is thus a direct comparison between Zeeman and thermal energies. The fraction of atoms $p(T, B)$ having enough collisional energy $E_C$ for a given temperature $T$ and an external magnetic field $B$ is given by

$$p(B, T) = \int_{\Delta E(B)/2}^{\infty} p(E_C) \, dE_C \,, \tag{2}$$

with $p(E_C)$ being the Maxwell-Boltzmann distribution of collision energies, resulting in

$$p(B, T) = 1 + \sqrt{\frac{\mu_{\mathrm{B}} B}{\pi k_{\mathrm{B}} T}} \exp\left(-\frac{\mu_{\mathrm{B}} B}{4 k_{\mathrm{B}} T}\right) - \mathrm{erf}\left(\sqrt{\frac{\mu_{\mathrm{B}} B}{4 k_{\mathrm{B}} T}}\right). \tag{3}$$

Hence, for high (low) temperatures and low (high) magnetic fields, the fraction of atoms capable of endoergic SE collisions is relatively high (low), as is illustrated by purple-shaded area in panels (a)-(d) of Fig. 2.

By contrast, exoergic SE collisions are always energetically allowed, converting single quanta of internal energy from Rb to Cs ($|F_{\mathrm{Cs}}, m_{F,\mathrm{Cs}}\rangle \rightarrow |F_{\mathrm{Cs}}, m_{F,\mathrm{Cs}} - 1\rangle$ and $|F_{\mathrm{Rb}}, m_{F,\mathrm{Rb}}\rangle \rightarrow |F_{\mathrm{Rb}}, m_{F,\mathrm{Rb}} + 1\rangle$). For simplicity, in the following, we write $m_{F,\mathrm{Cs}}$ as $m_F$.

## 3 Numerical model

In order to numerically model the outcome of a quantum probing result, we infer the spin dynamics by solving rate equations for the population transfer between different $m_F$-states. Coherences or off-diagonal elements in the spin-transfer matrix are neglected because the frequent elastic collisions will quickly dephase the coherence between two atoms in a SE collision before the next SE collision occurs. The rate of a spin exchange event is given by

$$\Gamma^{m_F} = \langle n \rangle \, \sigma_{m_F} \, \bar{v} \,. \tag{4}$$

Here

$$\langle n \rangle = \int n_{\mathrm{Cs}}(\vec{r}) \, n_{\mathrm{Rb}}(\vec{r}) \, d\vec{r} \tag{5}$$

denotes the Cs-Rb density overlap, $\sigma_{m_F}$ is the scattering cross section of the corresponding states ($m_F \rightarrow m_F + 1$ for endoergic collisions, and vice versa for exoergic SE events),

$$\bar{v} = \sqrt{\frac{8 k_{\mathrm{B}} T}{\pi \mu}} \tag{6}$$

is the relative velocity of the colliding atoms, and $n_{\mathrm{Cs}}(\vec{r})$ ($n_{\mathrm{Rb}}(\vec{r})$) the Cs (Rb) density. The scattering cross sections are results from a coupled channel calculation, matching the experimental observations in the parameter range used to a percent level [19]. Averaging the crossing sections for each $m_F$ transition over a Maxwell Boltzmann distribution yields twelve rates, six endoergic and six exoergic SE rates, in the seven-level system. The interaction-induced spin dynamic is determined using a differential equation including the rates $\Gamma^{m_F}, \Gamma^{m_F \pm 1}$ and state

population $P_{m_F}, P_{m_F \pm 1}$ of one state and its direct neighbor states, assuming that only collisions exchanging one quantum are possible. Starting from a given initial population distribution $P_{m_F}$ of the probe, the spin dynamics can hence be predicted by solving the differential equation

$$
\dot{P}_{m_F} = \begin{pmatrix} 0 & -\Gamma^{m_F \to m_F+1} & 0 & \\ \Gamma^{m_F+1 \to m_F} & 0 & \Gamma^{m_F-1 \to m_F} & \\ 0 & -\Gamma^{m_F \to m_F-1} & 0 & \cdots \\ & & \vdots & \end{pmatrix} \cdot \begin{pmatrix} P_{m_F+1} \\ P_{m_F} \\ P_{m_F-1} \\ \vdots \end{pmatrix}. \tag{7}
$$

The choice of the bath state $|F_{Rb} = 1, m_{F,Rb} = 0\rangle$ forbids collisions that exchange two quanta of angular momentum due to angular momentum conservation. Moreover, we assume that Rb atoms collide only once with Cs caused by the massive imbalance between Rb and Cs atom numbers $N_{Rb}/N_{Cs} \approx 1000$. Numerically solving equation 7 yields the spin dynamics as well as the steady state, which is used to compute the sensitivity of the probe. The dependence of the individual rates for SE processes on the internal state, but also the total energy $E_{tot}$ and the energy ratio $E_{ratio}$ (see Fig. 3), allows to deduce bath information from the steady state and also from the nonequilibrium spin dynamics even after few SE collisions have taken place.

## 4 Sensitivity

In this work, we refer to sensitivity as the change of a measurement outcome (here, the Cs quantum state population) for a given change in the observable of interest (here $E_{th}$, $E_Z$, $E_{th}/E_Z$, and $E_{tot}$). To determine the sensitivity of our system, we compare the simulated steady-state spin distributions for small parameter changes $\delta\theta$ in the parameter of interest $\theta$. This is quantified by, first, calculating the Bures distance $d_{Bures}$, given by [18, 20]

$$
d_{Bures}^2(\delta\theta) = 2 - 2 \sum_{m_F} \left[ P_{m_F}(\theta) P_{m_F}(\theta + \delta\theta) \right]^{1/2}. \tag{8}
$$

The Bures distance coincides for the case here with the Hellinger distance [21] because the probe's density matrix is quickly reduced to diagonal form, i.e., populations only, while the coherences are depleted by frequent elastic collisions between two SE collisions. Intuitively, the Bures distance quantifies the difference between two probe quantum states as a function of the parameter of interest $\theta \in [E_{th}, E_Z, E_{th}/E_Z, E_{tot}]$. A high sensitivity is signaled if a small change in the parameter of interest $\delta\theta$ results in a significant change in $d_{Bures}$.

This requirement is captured by the statistical speed $s$ [22] which we extract from a Taylor expansion for small values around the reference point (where $d_{Bures}(\delta\theta = 0) = 0$) to first order of the Bures distance

$$
s(\delta\theta = 0) = \frac{\partial d_{Bures}}{\partial \delta\theta} = \sqrt{\frac{F_\theta(\delta\theta = 0)}{8}}. \tag{9}
$$

This equation relates the statistical speed to the square root of the Fisher information, which we use as sensitivity. The first-order Taylor expansion properly describes the Bures distance behavior around the reference point (zero point), as shown as dashed lines in Fig. 4.

We investigate the sensitivity with respect to different energy contributions, as shown in Fig. 2, where two orthogonal axes of an energy plane are spanned by thermal $E_{th}$ and Zeeman $E_Z$ energies. The diagonal thus represents the total energy $E_{tot}$, while the orthogonal "anti-diagonal" corresponds to the energy ratio $E_{ratio}$ for given total energy. Only atoms with sufficient (thermal) collision energy ($E_C \geq \Delta E/2$) can undergo an endothermal SE collision.

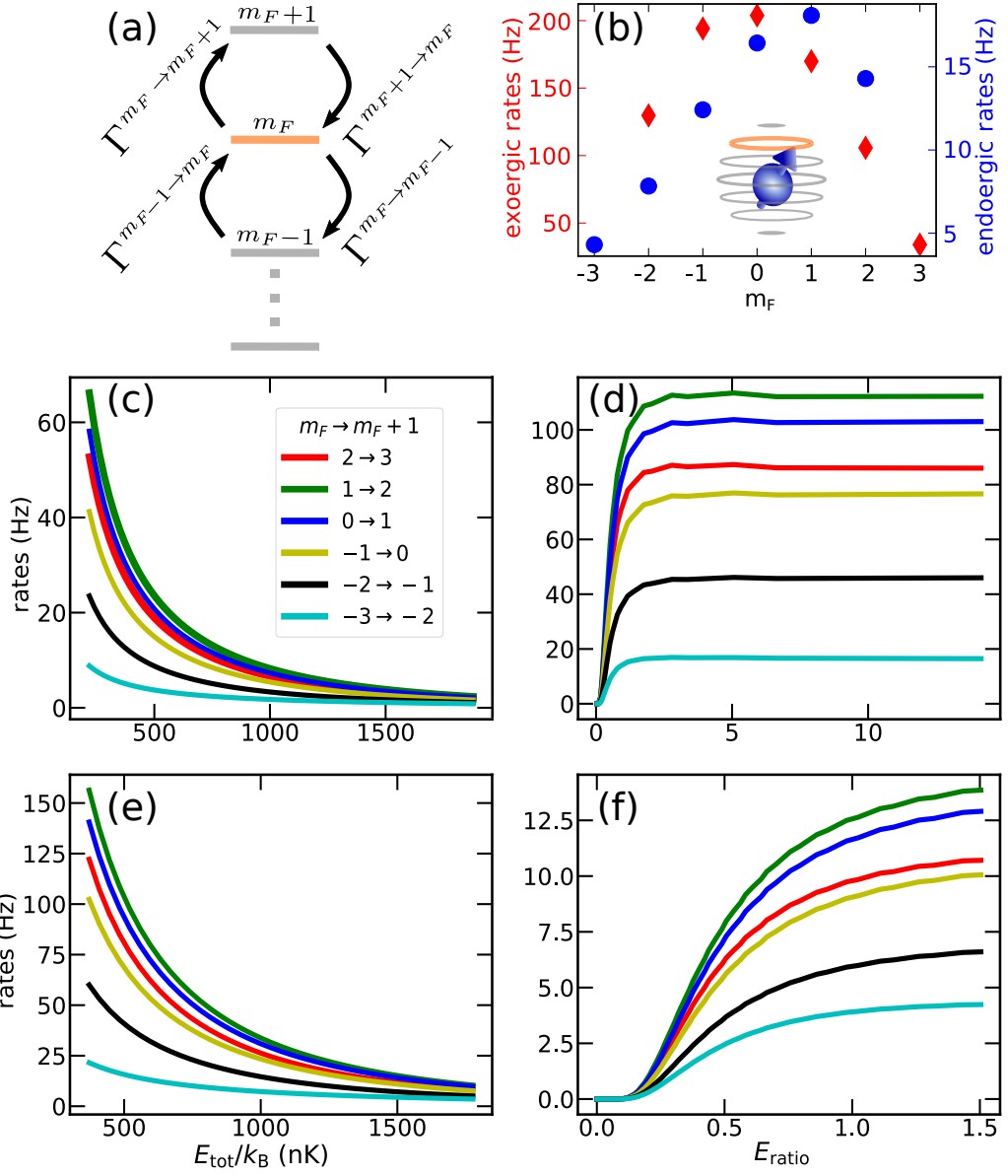

Figure 3: **Spin-exchange model.** For clarity, only rates between three of seven Cs states are shown in (a), resulting in a differential equation. Endoergic (blue dots) and exoergic (red diamonds) rates are shown exemplarily for $B = 43\,\text{mG}$ and $T = 435\,\text{nK}$ (b), which are typical values of the experiment. Total Cs spin space is illustrated as Bloch sphere in the inset. (c)-(f) present the rate dependency on the total energy for a fixed energy ratio (c) $E_{\text{ratio}} = 0.3$, (e) $E_{\text{ratio}} = 1.2$, and on the energy ratio for fixed total energy (d) $E_{\text{tot}}/k_{\text{B}} = 0.7\,\mu\text{K}$, (f) $E_{\text{tot}}/k_{\text{B}} = 2.2\,\mu\text{K}$. Colors in (c)-(f) label the different endoergic spin transfer rates for different state transitions $|m_F\rangle \rightarrow |m_F + 1\rangle$.

Thus, these collision mechanisms primarily mediate the information contained in the single-atom probe, which is measurable via the probe's spin state, for more details see [1]. In the following, we numerically study the sensitivity along all four axes and relate it to the form of the fraction of endoergic SE processes $p(B, T)$ given in eq. (3).

## 5  Comparison of sensing applications

To infer the sensitivity for all four directions of the energy plane in Fig. 2, representing the four sensing applications of calorimetry ($E_{\text{tot}}$), thermometry ($E_{\text{th}}/k_B$), magnetometry ($E_Z/\mu_B$), and the ratio of energy contributions ($E_{\text{ratio}}$), we numerically compute the Bures distance and subsequently the Fisher information from eqs. (8,9) at the same reference state $P_{m_F}(\theta)$ at $T = 435\,\text{nK}$ and $B = 43\,\text{mG}$ (marked as diamonds in Fig. 2 and Fig. 4). The temperature is chosen with respect to typical values reached in our experiment and is thus an ideal starting point.

The resulting Bures distance along the four axes of interest are shown in Fig. 4. It shows that, in all directions, the Bures distance changes approximately linearly around the reference point. The obvious asymmetry of the Bures distance $d_{\text{Bures}}$ with respect to the reference point results from an asymmetric energy condition for endoergic collision mechanisms and the differing collision rates of endo- and exoergic processes. To account for this asymmetry, we heuristically assume that the concept of the Fisher information independently holds in each direction in parameter space and compute the Fisher information using two Taylor expansions,

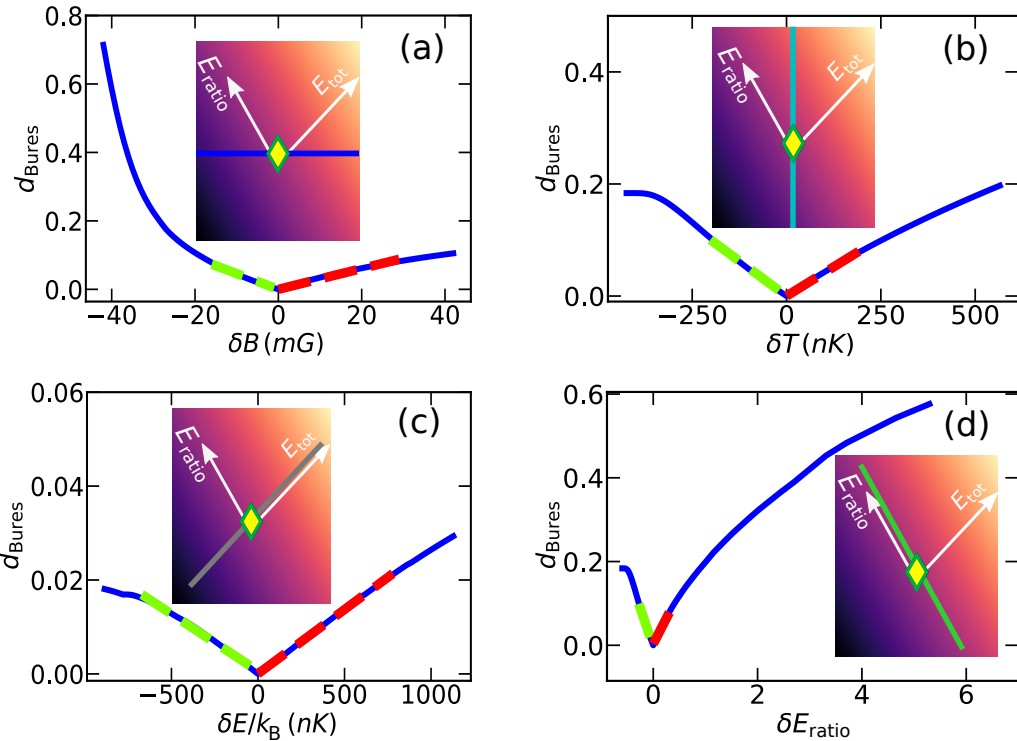

Figure 4: **Bures distances.** For constant temperature $T = 435\,\text{nK}$ (a), a constant magnetic field $B = 43\,\text{mG}$ (b), a constant energy ratio $E_{\text{ratio}} = 0.6$ (c) and for a constant total energy $E_{\text{tot}}/k_B = 1.6\,\mu\text{K}$ (d). The reference state at $T = 435\,\text{nK}$ and $B = 43\,\text{mG}$ is marked by the diamond as in Fig. 2. The dashed lines represent the Taylor expansion to the first order.

one for each side of $d_{\mathrm{Bures}}$ around the reference point (positive and negative $\delta\theta$). The Bures distance is zero when the reference state $P_{m_F}(\theta)$ and the comparison state $P_{m_F}(\theta + \delta\theta)$ in Eq. 8 are equal. We observe that the Bures distance varies by more than one order of magnitude for the different axes, indicating already a strongly differing sensitivity along the different directions. Specifically, we observe an increasing Bures distance for lower magnetic fields in Fig. 4(a), which we explain by a strongly increasing fraction of endoergic SE processes for small magnetic fields. The variation of the temperature at a fixed magnetic field Fig. 4(b) reveals an almost constant slope for a broad range of parameters. Fixing the energy ratio $E_{\mathrm{ratio}} = 0.6$ yields a constant endoergic SE fraction where the slope is significantly flatter compared to the other cases, see Fig. 4(c). Finally, the greatest substantial variation in the Bures distance occurs when the total energy is fixed (here $E_{\mathrm{tot}}/k_{\mathrm{B}} = 1.6\,\mu\mathrm{K}$), altering the energy ratio Fig. 4(d).

Concluding, for the reference point given, the system is most sensitive along the axis of the energy ratio and least sensitive along the axis of the total energy, where the endoergic SE fraction is nearly constant. Thus, the microscopic mechanisms render the single-atom probe an excellent energy balance, a decent thermometer or magnetometer, and a poor calorimeter.

## 6 Points of maximum sensitivity

The SE rates depend on various external parameters, concretely temperature and external magnetic field, as shown in Eq. (4). It is also expected that the sensitivity depends on the specific choice of the reference state. This is particularly important if an unknown many-body system is to be probed and the optimal probing strategy is searched for.

We therefore now address the question of how the reference state influences the sensitivity, and at which reference point the sensitivity can be maximized. To investigate the sensitivity behavior along the four directions that pass the center in Fig. 2 and extract the maximum sensitivity, we change the reference state $P_{m_F}(\theta)$ in Eq. (8) along each direction and extract $\sqrt{F_\theta}$.

Fig. 5 shows a maximum for most sensing applications, where for the case of constant temperature in (a) the maximum at ultra-low magnetic fields is not considered in this work because it is experimentally not controllable. The general behavior of the sensitivity curves can be understood as follows. For a constant temperature, the observed decrease in sensitivity with larger magnetic fields Fig. 5(a) reflects the decrease in endoergic SE events. For constant magnetic field, Fig. 5(b), endothermal collisions are absent for very small temperatures and as probable as the exothermal collisions for high temperatures. In both limiting cases, a small change in temperature does not yield a change of the steady state spin distribution. This suggests, however, a sensitivity maximum somewhere for intermediate temperatures. When the energy ratio $E_{\mathrm{th}}/E_Z$ is left unchanged Fig. 5(c), the sensitivity changes only slightly compared with the magnitude of the other cases, because the relative contribution of endoergic SE fraction to the SE collisions does not change significantly.

As suggested by the different scales associated to the Bures distance, the four instances have clearly different degrees of sensitivity, and the probe is most sensitive when the energy ratio $E_{\mathrm{th}}/E_Z$ is changed Fig. 5(d). In the following, we aim at understanding the origin of the maximum in sensitivity from studying the microscopic mechanism and specifically the endothermal SE collisions. To this end, we focus on the case of probing the energy ratio between thermal and Zeeman energies.

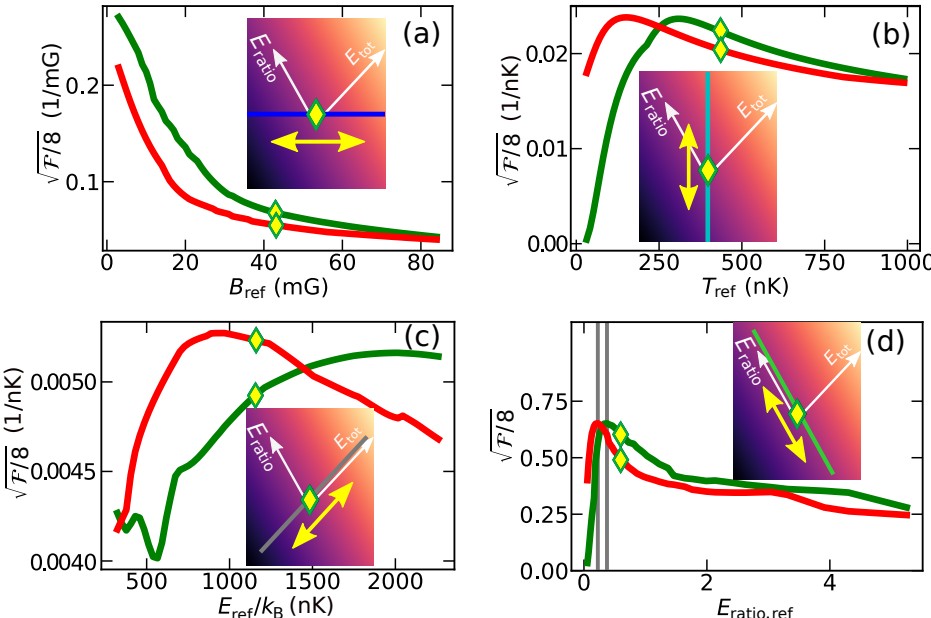

Figure 5: **Sensitivity for a variation of reference states.** The reference state $P_{m_F}(\theta)$ (diamonds in the insets) is varied for constant temperature $T = 493$ nK (a), a constant magnetic field $B = 43$ mG (b), a constant energy ratio $E_{\text{ratio}} = 0.6$ (c) and a constant total energy $E_{\text{tot}}/k_B = 1.6\,\mu$K (d) indicated by the double arrows. Green (red) line represents the sensitivity for the left, i.e., $\theta < 0$ (right, i.e., $\theta > 0$) side of the Bures distance. Sensitivity determined from Fig. 4 is marked by diamonds in the main graphs. Vertical lines mark maxima, investigated in detail in the following.

# 7 Experimental realization

We immerse single $^{133}$Cs atoms into an ultracold $^{87}$Rb bath, consisting of $N = 5\ldots9 \times 10^3$ Rb atoms at densities of $10^{12}\ldots10^{13}$ cm$^{-3}$ and temperatures in a range of $T = 0.2 - 1\,\mu$K in the $|F_{\text{Rb}} = 1, m_{F,\text{Rb}} = 0\rangle$ state. Cs is prepared 160 $\mu$m away from the Rb cloud center in the state $|F_{\text{Cs}} = 3, m_{F,\text{Cs}} = 3\rangle$. Microwave Landau-Zener transitions prepare the degenerated Raman sideband-cooled Cs [23] in the state $|F_{\text{Cs}} = 3, m_{F,\text{Cs}} = 2\rangle$. Subsequently, the Cs atoms are transported into the Rb cloud by guiding them along the joint axial trapping potential.

Immersed in the Rb cloud, the Cs atoms' kinetic state quickly thermalizes to the Rb bath temperature (after approx. three elastic collisions) before the first spin-exchange collision takes place. For the parameters considered in this work, spin-exchange collisions are less frequent by a factor of approximately ten, so that the Cs atom can be considered thermalized for each SE collision.

After the interaction, a series of microwave transitions at a frequency of 9.1 GHz promotes selected Cs $m_F$ states to the $F = 4$ manifold, and a pushout laser pulse resonantly excites the $F = 4$ population, removing them from the trap. Repeating this for different Cs $m_F$ states allows to resolve the Cs spin populations as a function of interaction time [16]. The external magnetic field during the interaction, ranging from $B = 10-80$ mG, is calibrated via microwave spectroscopy of the Rb cloud on the $|F_{\text{Rb}} = 1, m_{F,\text{Rb}} = 0\rangle \rightarrow |F_{\text{Rb}} = 2, m_{F,\text{Rb}} = 1\rangle$ Rb microwave transition.

In order to compare the findings of our numerical investigations with experimental data, we have recorded spin dynamics of single-atom probes immersed in an ultracold Rb gas in a wide range of accessible temperature and magnetic field values, indicated as data points in

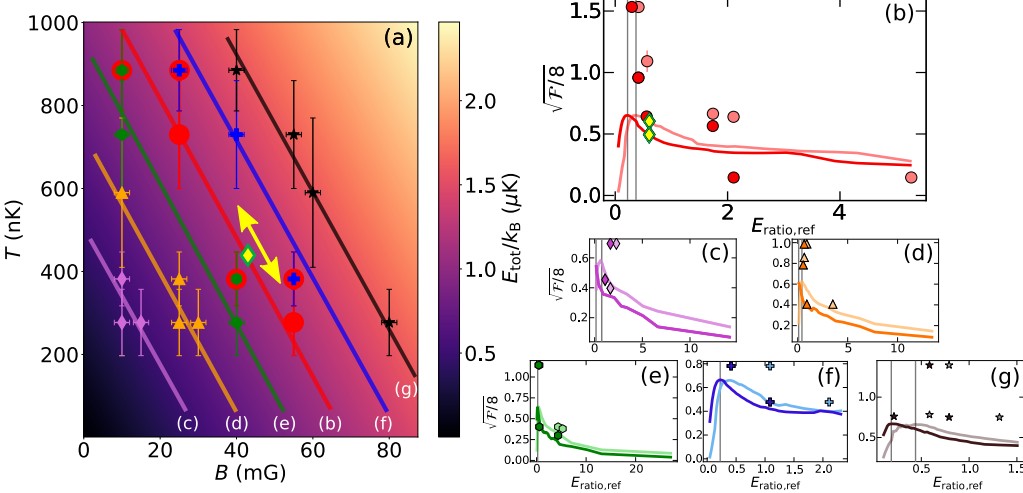

Figure 6: **Data and simulation regimes for sensitivity determination in parameter space.** (a) Parameter range for experimentally obtained data and simulations. The sensitivities have been calculated for different total energies, where the yellow double arrow illustrates the change of the reference state along the lines (for simulation) and the grouped data points. Due to experimental drifts over long times, the data shows deviations and uncertainties mainly in temperature. We therefore group data points into pre-defined total energies (indicated by symbols) with an average deviation of less than 7% to the corresponding simulated total energy. The lines indicate the parameters used for corresponding simulations. The magnetic-field error $\Delta B = 2\,\mathrm{mG}$ is assumed to be constant. For more details see [1]. Temperature errors reflect shot to shot fluctuation. Colorbar gives the total energy. Comparison of measured sensitivity from nonequilibrium data (data points) and steady-state simulations (lines) for the parameter indicated by symbols and colors in (a). Simulation is calculated for $E_{\mathrm{tot}}/k_{\mathrm{B}} = 1.6\,\mu\mathrm{K}$ (red lines (b)), $0.7\,\mu\mathrm{K}$ (pink lines (c)), $1.1\,\mu\mathrm{K}$ (orange lines (d)), $1.3\,\mu\mathrm{K}$ (green lines (e)), $1.91\,\mu\mathrm{K}$ (blue lines (f)) and $2.2\,\mu\mathrm{K}$ (black lines (g)). Faded colored (intense colored) lines and data points give the sensitivity of the linear part left (right) of the Bures distance. Diamonds in (a) and (b) mark the reference state (at $T = 435\,\mathrm{nK}$ and $B = 43\,\mathrm{mG}$ for a total energy of $1.6\,\mu\mathrm{K}$) considered in Fig. 4(d), vertical lines mark the maxima of the two simulated curves.

Fig. 6(a). The comparison with numerics is complicated by the fact that, for many combinations of thermal and Zeeman energies, the life time of the single-atom probe is too low to experimentally reach the steady state. Since we are interested in finding the energy ratio of the maxium sensitivity rather than agreement in the absolute value of the sensitivity, we compare the steady state simulation to nonequilibrium data, extracting population distributions, Bures distances and statistical speed from the measured spin dynamics, see [1]. As shown there, the sensitivity based on the nonequilibrium spin dynamics is significant larger than the one using the steady state. The direct comparison will therefore shows quantitative differences of the sensitivity, but the steady-state simulations predict the positions of the sensitivity maxima. The data sets show a scatter in parameter space, where temperature and external magnetic field have been independently determined. We group them into six pre-defined total energies, indicated by colored lines in Fig. 6(a) and assign them to a group if the difference in total energy is smaller than 25% (average deviation is less than 7%) indicated bys the corresponding colored filling or frame of the data point. Thereby, some experimental data sets can contribute to two groups of different total energy.

To extract the sensitivity, we analyze the data similar to the numerical investigations using Eqs. (8) and (9). However, the Bures distances here are calculated purely from different experimentally recorded Cs populations and contain no numerical model. We note that, strictly, this is not a differential measurement as required by Eq. (9) to compute the statistical speed. However, for most combinations, neighboring data sets lie within the parameter range where the Bures distance still scales linearly, and we plot the resulting statistical speed in Fig. 6(b)-(g). For data sets with large distance in parameter space in Fig. 6, for example the data shown in (f) (blue) or (g) (black), this might not be fulfilled, leading to an enhanced deviation from the prediction.

For each total energy, we additionally plot the numerically expected steady-state sensitivity as solid line in Fig. 6(b)-(g), and indicate the position of the inflection points of the probability distribution for endothermal SE collisions as vertical solid lines. As expected, numerical and experimental data differ quantitatively in the magnitude of sensitivity. The vertical solid lines for all data sets coincide with the maxima of the numerically predicted sensitivity. Moreover, for sufficient dense data in parameter space and small total energies, we observe for the nonequilibrium experimental data that the sensitivity maxima are also close to the vertical lines, irrespective of the magnitude of the sensitivity.

We deduce from this observation that, first, for a broad range in parameter space, the sensitivity shows a nonmonotonic behavior. We emphasize that the sensitivity maxima observed here in parameter space are different from the ones observed in the nonequilibrium time evolution. Second, a discrete measurement of the sensitivity is possible with suprisingly large distances in parameter space and, hence, Bures distances, as the statistical speed is constant in a relatively large range of parameters.

## 8 Microscopic origin of maximum sensitivity

As illustrated in Fig. 2, microscopically, the probe information is predominantly mediated by endothermal SE processes, which are determined by a competition of Zeeman and thermal collision energies. We therefore study more closely the fraction of collisions having sufficient kinetic energy to promote an endothermal SE event, given by Eq. (3). Fig. 7 shows this endoergic SE fraction as a function of the energy ratio $E_{\mathrm{th}}/E_{\mathrm{Z}}$, together with its first and second derivatives.

The fraction of endothermal SE increases with larger thermal energy and/or lower Zeeman energy. Importantly, from our numerical investigations of the sensitivity, we find that the maximum sensitivity for the left (right) wing of $d_{\mathrm{Bures}}$ corresponds to energy ratios close to the maximum of the first (second) derivative. This can be intuitively understood from simple arguments. The probe atom's spin distribution is driven in opposite directions ($m_F = \pm 3$) by exoergic and endoergic SE. Differing collision rates between the two collision processes and a relatively strong change of the endoergic SE fraction at the inflection points lead to a significant shift in the system's steady state. As a result, it is most sensitive in this regime. This intuitive explanation for the origin and position of the sensitivity maxima based on the inflection points of the endothermal collision probability distribution yields good predictions in Fig. 6 also for dynamics out of equilibrium, while the previously known differences in absolute sensitivity between equilibrium and nonequilibrium dynamics persist for all parameters. The connection between the fraction of endothermal SE and points of maximum sensitivity for other constant total energies in Fig. 6 are depicted in the appendix in Fig. 8.

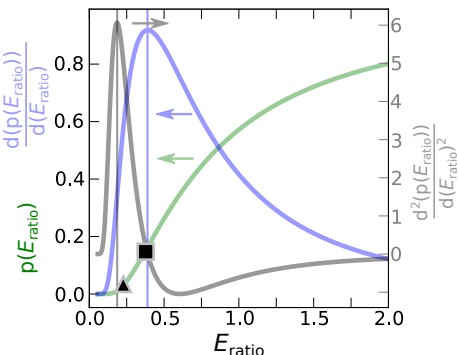

Figure 7: **Endoergic SE fraction** $p(E_{\text{ratio}})$ **along the direction of constant energy for** $E_{\text{tot}}/k_{\text{B}} = 1.6\,\mu\text{K}$. The green line represents the fit function $f(E_{\text{ratio}}) = -1.29/(1 - 2.29e^{0.35x^{-1.43}})$ of the endoergic fraction along the constant energy axis calculated by Eq. 3. Blue and grey lines show the first and second derivatives of the fit function. The maximum sensitivity at an endoergic SE fraction of 0.15 (0.03) for the left (right) side of the $d_{\text{Bures}}$ is illustrated as a black square and triangle. The vertical lines mark the maxima of the derivatives.

## 9  Conclusion

The observation of sensitivity maxima in parameter space for equilibrium states pave the way for optimization of such collisional quantum probes in parameter space beyond the previously known nonequilibrium boost. Our considerations of the probability distribution of endothermal collisions connect measures for the probe's sensitivity with the microscopic mechanism. While the absolute sensitivity changes out of equilibrium, the maximum in $(B, T)$ parameter space can be expected to be close to the value of the equilibrium case. A particularly interesting option for optimization in this context arises as the information deduced from the probe comes in individual quanta from each measurement. It will be interesting in the future to optimize the probing strategy by balancing the limited gain of information flow for unkown reference states in $(B, T)$ parameter space on the one hand, with the nonequilibrium boost on the other hand. Our work shows that an optimum reference point exists and can be found with increasing knowledge, hence measurement time. The nonequilibrium boost by contrast assumes perfect knowledge of the reference point but shows best performance for short interaction times. An optimal strategy must balance between both mechanisms with competing requirements for interaction time.

The experimental platform together with the level of understanding throughout the parameter space provided in this work might make our system interesting for testing future concepts and scenarios of quantum probing. A first example is to investigate memory effects of the bath. All considerations so far assume that the Rb bath is Markovian, i.e., it does not retain any memory of the probe-bath interaction. In our experiment, this is justified, because the probe-bath interaction leads to few Rb atoms in a different spin state, and the probability of the impurity to collide again with these atoms is negligible. Reducing the size of the bath by reducing the number of Rb atoms, however, will allow realizing a situation where this probability of re-colliding between probe and one specific bath atom for a second time becomes relevant. This will allow us studying the effect of memory and bath correlations onto the performance of quantum probing. A second example might be to spark further work elucidating the consequences of the different statistical speeds occurring for some parameter combinations. Finally, it will be interesting to compare the collisional quantum probing, perturbing the bath by individual quanta of angular momentum through the SE collisions, with single-atom coherent

probes [24], where information about the gas is mapped onto the quantum superposition of two internal probe states.

## Acknowledgements

**Funding information** This work was funded by Deutsche Forschungsgemeinschaft (DFG) via Sonderforschungsbereich SFB/TRR 185 (Project 277625399).

## A Appendix

Fig.8 depicts the behavior of the endoergic SE fraction and their first two derivatives for all constant total energies presented in Fig. 6. The behavior is qualitatively the same as in Fig. 7. Especially for higher total energies ($E_{tot}/k_B > 0.7\,\mu K$), consistency of the points of maximum sensitivity can be seen by the correspondence of these points to the first (second) derivative.

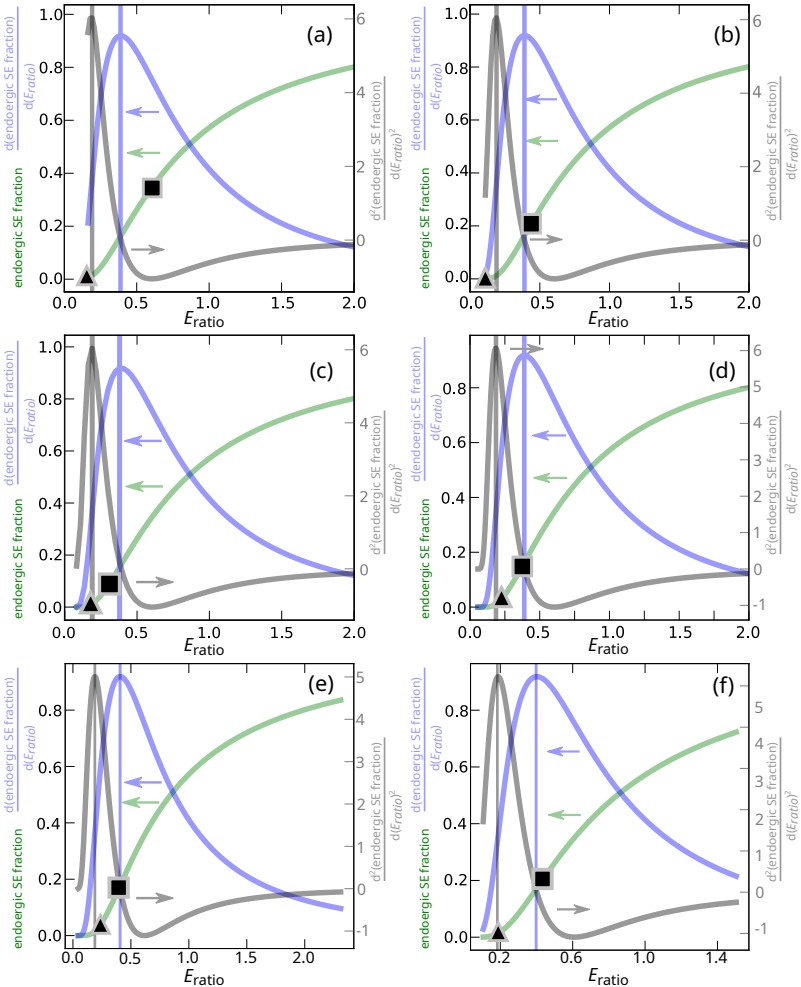

Figure 8: **Endoergic SE fraction along the direction of constant energy for different $E_{tot}$.** (a) $E_{tot}/k_B = 0.7\,\mu K$, (b) $1.1\,\mu K$, (c) $1.3\,\mu K$, (d) $1.6\,\mu K$, (e) $1.91\,\mu K$ and (f) $2.2\,\mu K$. The green line represents the fit function of the endoergic fraction along the constant energy axis calculated by Eq. 3. Blue and dark lines show the first and second derivatives of the fit function. The maximum sensitivity for the left (right) side of the $d_{Bures}$ is illustrated as black squares and triangles, as in Fig. 2. The vertical lines mark the maxima of the derivatives.

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
