# Peer review of "Sensitivity of a collisional single-atom spin probe"

_SciPost Physics, doi:SciPost Phys. Core 6, 009 (2023)_

## Round 1 · Referee Report · Anonymous (Referee 4) · 2022-10-18

Report

I have evaluated the revised version of the paper

Regarding my comment on the relevance of the findings in comparison to the existing literature, the authors clarify what constitutes novelty in the present manuscript.

Regarding the comparison to experiments, the authors agree, stating it clearly in the text, that a direct comparison is not provided due to significant differences.

The new conclusions are well written and highlight the relevance of the paper.

Overall, I don't think that the paper meets the relevance criteria of SciPost. In my opinion, it would be suitable for publication in Physical Review A.

  • validity: -
  • significance: -
  • originality: -
  • clarity: -
  • formatting: -
  • grammar: -

Author:  Jens Nettersheim  on 2022-11-13  [id 3011]

(in reply to Report 1 on 2022-10-18)

Referee: Overall, I don't think that the paper meets the relevance criteria of SciPost. In my opinion, it would be suitable for publication in Physical Review A.

Authors: We thank the referee for his/her opinion.

---

## Round 1 · Referee Report · Anonymous (Referee 5) · 2022-10-23

Report

The authors have answered all my questions and made suitable changes to the manuscript.

Regarding the impact of this work, which I questioned in my previous report, the authors reply:

"We believe that, in view of our modifications in the revised version, the manuscript meets acceptance criterion 3 (Open a new pathway in an existing or a new research direction, with clear potential for multipronged follow-up work). While the work in Ref. [15] indicated that the quantum probe operation can be optimized in time, the present work shows that optimization in parameter space (B,T) or, equivalently, total energy and energy ratio, independent of time is additionally available. "

I have a different opinion on this matter, as I don't recognize sufficient innovation to "Open a new pathway in an existing or a new research direction, with clear potential for multipronged follow-up work)". This paper demonstrates that an equilibrium analysis captures the main features of a carefully modelled nonequilibrium analysis, which was previously published. These results are worth being published in a peer-reviewed journal without any doubt, but appear more of the incremental type rather than opening a new path or research direction.

After reading the new manuscript, I collected new comments for the authors, which may help improve the presentation:

The authors have followed the recommendation (which was also given by the other referees) to improve the presentation of the experimental results by rewriting and extending Section 8.

The new Section 8 has clearly improved with the respect to the previous version. However, a criticism still remains that the two sections of the paper, the theoretical and experimental one, are not harmonized together. In the current version, the experimental section reads as an appendix attached to the theoretical paper. The paper would greatly profit if it would be structured to have the experimental results presented before Sec. 6 "Points of maximum sensitivity" and Sec. 7 "Microscopic origin of maximum sensitivity". This change should be possibe to be implemented with minor effort. With the proposed change, the sections currently labelled 6 and 7 will then provide an insight and interpretation of the observed experimental results, which are now presented in Sec. 8.

A second point of criticism concerns the caption of Figure 7. While Section 8 has been largely rewritten, the caption of the associated Fig. 7 has not been sufficiently revised and remains unclear and may contain inaccuracies. In this regard, please refer to the following comments:

-1 The caption begins with "Temperatures, magnetic fields and total energies of data and the corresponding simulation (left) and their sensitivities (right panels (a)-(f))". First, there seem to be a mistake where panel (a)-(f) should presumably be (b)-(g). Second, where are the simulations in the panel on the left-hand side? To my understanding, equilibrium simulations/calculations are provided for panels (b)-(g), but not panel (a). Moreover, in the interest of clarity, it is advisable to avoid speaking of "left" and "right," but rather mention the relevant panels explicitly.

-2 The rest of the caption is also very difficult to follow. For example, the different panels are listed in a noncanonical order: (c), (d), (e), (b), (f), (g). More over and more importantly, the sentence only speaks of "Simulation are calculated for E_tot = ..." without mentioning data but then in the parentheses some additional references are provided (diamonds, triangles, etc), which — I presume — are experimental data. The sentence should be reformulated to mention the experimental data as well.

-3 Where in the caption it is written "considered before Fig. 4, vertical lines mark the maxima.", does it mean "considered in Fig.4" instead of "before Fig.4"?

-4 Do the diamond points in the figure 7(a) (as well as in Fig.5) denote the reference points with respect to which to compute the discrete difference, as an approximation for the derivatives needed in the definition of the Fisher information? If so, how can the finite difference be computed for the reference points? Related to this point, it is advisable to be very explicit how exactly the finite difference is evaluated to obtain an approximation of the statistical speed (i.e., sqrt of the Fisher information). I am still confused about what exactly is plotted in Fig. 7(b-g), since the linear behavior of the Bures distance, visible in Fig. 4, only applies in the surrounding of the reference point. When moving far from it, the behavior is clearly no longer linear, and a different method other than discrete differences should be used to evaluate the derivates.

-5 In the caption of Fig.7, "vertical lines mark the maxima." should say of what. I presume of the two simulated curves.

The authors have answered all my questions and made appropriate changes to the manuscript in most of the cases.

The authors have followed my recommendation (the same also made by the other reviewers) to improve the presentation of the experimental results by rewriting and expanding Section 8. The new Section 8 has now improved significantly over the previous version. However, one criticism remains that the two sections of the paper, the theoretical and the experimental, do not harmonize. As it stands, the experimental section reads like an appendix to the theoretical section. The paper would benefit greatly if the experimental results were presented before Sec. 6 "Points of maximum sensitivity" and Sec. 7 "Microscopic origin of maximum sensitivity". This change should be implementable with little effort. With the proposed change, the sections currently labeled 6 and 7 will then provide insight and interpretation of the observed experimental results now presented in Section 8.

A second criticism concerns the caption of Figure 7. While Section 8 has been largely rewritten, the caption of the associated Figure 7 has not been sufficiently revised and remains unclear and may contain inaccuracies. In this regard, I refer to the following comments:

-1 The figure caption begins with "Temperatures, magnetic fields and total energies of data and the corresponding simulation (left) and their sensitivities (right panels (a)-(f))". First, an error seems to have slipped in, as panels (a)-(f) should probably be (b)-(g). Second, where are the simulations in the left panel? As far as I know, equilibrium simulations are given for panels (b)-(g), but not for panel (a) on the left. Also, in the interest of clarity, it is advisable not to talk about "left" and "right", but to explicitly mention the corresponding panels.

-2 The rest of the caption is also very difficult to understand. For example, the various panels are listed in a non-canonical order: (c), (d), (e), (b), (f), (g). Furthermore, and more importantly, the sentence only talks about "Simulation are calculated for E_tot = ..." without mentioning any data, but then giving some additional references in the parentheses (diamonds, triangles, etc.), which I assume are experimental data. The sentence should be reworded to mention experimental data as well.

-3 When the caption says "considered before Fig. 4, vertical lines mark the maxima.", does that mean "considered in Fig.4" instead of "before Fig.4"?

-4 Do the diamond points in Fig. 7(a) (as well as in Fig. 5) denote the reference points with respect to which the discrete difference is calculated, as an approximation for the derivatives needed in the definition of the Fisher information? If so, how can the finite difference be calculated for the reference points? Related to this point, it is advisable to be very specific in the main text about how the finite difference is evaluated to get an approximation to the statistical velocity (i.e. sqrt of Fisher information). This is a question I had in my previous report. I am still confused about what exactly is plotted in Fig. 7(b-g), since the linear behavior of the Bures distance seen in Fig. 4 holds only in the vicinity of the reference point. Moving far away from it, the behavior is clearly no longer linear, and a method other than discrete differences should be used to evaluate the derivatives for those points far from the reference.

-5 The caption of Fig. 7 says "vertical lines mark the maxima", but it is not clear of what? I presume these are the maxima of the two simulated curves.

  • validity: good
  • significance: ok
  • originality: ok
  • clarity: ok
  • formatting: reasonable
  • grammar: good

Author:  Jens Nettersheim  on 2022-11-13  [id 3012]

(in reply to Report 2 on 2022-10-23)

Referee: The new Section 8 has clearly improved with the respect to the previous version. However, a criticism still remains that the two sections of the paper, the theoretical and experimental one, are not harmonized together. In the current version, the experimental section reads as an appendix attached to the theoretical paper. The paper would greatly profit if it would be structured to have the experimental results presented before Sec. 6 "Points of maximum sensitivity" and Sec. 7 "Microscopic origin of maximum sensitivity". This change should be possibe to be implemented with minor effort. With the proposed change, the sections currently labelled 6 and 7 will then provide an insight and interpretation of the observed experimental results, which are now presented in Sec. 8.

Authors: The current structure helps to understand. After calculating the Bures distance of all four directions, the corresponding sensitivities are extracted. After understanding the sensitivity maxima origin, we extend the manuscript to data. Moving Section 8 before Section 6 would destroy this logical structure, especially Sections 5 & 6. Moving Section 8 before Section 7 would be reasonable, in our opinion. We moved section 8 before section 7 and adapted relevant text passages.
#########################

Referee: A second point of criticism concerns the caption of Figure 7. While Section 8 has been largely rewritten, the caption of the associated Fig. 7 has not been sufficiently revised and remains unclear and may contain inaccuracies. In this regard, please refer to the following comments:
-1 The caption begins with "Temperatures, magnetic fields and total energies of data and the corresponding simulation (left) and their sensitivities (right panels (a)-(f))". First, there seem to be a mistake where panel (a)-(f) should presumably be (b)-(g). Second, where are the simulations in the panel on the left-hand side? To my understanding, equilibrium simulations/calculations are provided for panels (b)-(g), but not panel (a). Moreover, in the interest of clarity, it is advisable to avoid speaking of "left" and "right," but rather mention the relevant panels explicitly.

Authors: First, we corrected the mistake in caption 7. Second, the reference states are changed along the lines in (a). The resulting sensitivity is shown in (b-g) as lines. Therefore (a) shows the direction the analysis is done. Lines give the analysis path for steady-state simulations, and symbols indicate the nonequilibrium data that are (symbol-wise) compared with each other (same analysis procedure for the simulations). We clarified this in the caption.
#########################

Referee: -2 The rest of the caption is also very difficult to follow. For example, the different panels are listed in a noncanonical order: (c), (d), (e), (b), (f), (g). Moreover and more importantly, the sentence only speaks of "Simulation are calculated for E_tot = ..." without mentioning data but then in the parentheses some additional references are provided (diamonds, triangles, etc), which — I presume — are experimental data. The sentence should be reformulated to mention the experimental data as well.

Authors: We ordered it alphabetically. We removed the symbols (diamonds, triangles, ect.) in parentheses and rewrote the sentence about the data points.
#########################

Referee: -3 Where in the caption it is written "considered before Fig. 4, vertical lines mark the maxima.", does it mean "considered in Fig.4" instead of "before Fig.4"?

Authors: We corrected this typo.
#########################

Referee: -4 Do the diamond points in the figure 7(a) (as well as in Fig.5) denote the reference points with respect to which to compute the discrete difference, as an approximation for the derivatives needed in the definition of the Fisher information? If so, how can the finite difference be computed for the reference points? Related to this point, it is advisable to be very explicit how exactly the finite difference is evaluated to obtain an approximation of the statistical speed (i.e., sqrt of the Fisher information). I am still confused about what exactly is plotted in Fig. 7(b-g), since the linear behavior of the Bures distance, visible in Fig. 4, only applies in the surrounding of the reference point. When moving far from it, the behavior is clearly no longer linear, and a different method other than discrete differences should be used to evaluate the derivates.

Authors: In Fig. 7 (now Fig. 6), the reference state along each line and along a set of data points is changed. The diamonds mark exemplarily one reference point. The Bures distance is shown in Fig. 4(d) for this reference point. The sensitivity of the diamonds is shown in Fig. 5(d). Infinitesimal small changes can not be resolved experimentally due to, for example, fluctuations. Therefore we extracted the sensitivity from the grouped data shown in Fig 7 (a).
#########################

Referee: -5 In the caption of Fig.7, "vertical lines mark the maxima." should say of what. I presume of the two simulated curves.

Authors: We clarified it in the caption.

---

## Round 1 · Referee Report · Anonymous (Referee 6) · 2022-10-24

Strengths

Experimental work, summary of experimental results on topic.

Weaknesses

Less weaknesses than before, now only weakness is the potential significance.

Report

Significant changes have ben made to improve the paper. I am only a bit astonished by the following comment "It is not the goal of this paper to establish the best agreement ..." which is quite a weak reply to my question/suggestion.

Requested changes

No specific requests.

  • validity: high
  • significance: good
  • originality: good
  • clarity: high
  • formatting: excellent
  • grammar: excellent

Author:  Jens Nettersheim  on 2022-11-13  [id 3013]

(in reply to Report 3 on 2022-10-24)

Referee: I am only a bit astonished by the following comment "It is not the goal of this paper to establish the best agreement ..." which is quite a weak reply to my question/suggestion.

Authors: It is our aim to provide agreement of simulation and experimental data of the relative energy where the maximum sensitivity occurs. This has been achieved. However, due to the different conditions of numerical simulation (steady state) and measured data (nonequilibrium), there is a discrepancy in the absolute value of the sensitivity, which is not relevant for determining the maximum. We have clarified this in the manuscript.

---

## Round 1 · List of Changes

List of changes:
-In the abstract, we clarified the novelty of our work compared to our previous work and the potential for future work.
-In the introduction, we clarified the results found in this work.
-In chapter 3, we comment on the condition for spin-exchange collisions that exchange two quanta of energy and the massive imbalance between the number of impurity and bath atoms.
-In chapter 4, we added that the Bures distance coincides for our case with the Hellinger distance and introduced the statistical speed that replaces equation (9). Moreover, we comment on the first-order Taylor expansion.
-In chapter 5, we mentioned that the concept of the Fisher information is based on a heuristic assumption.
-In chapter 8, we changed some text to clarify how the impurity is transported into the bath and added text for describing Fig. 7. We finished this chapter by summarizing the novel points found in this work.
-We rewrote the conclusion.
Figures:
- Figure 3: We corrected the typo in the legend.
- Figure 4: We removed the units of the y-axes.
- Figures 5, 7: We changed the y-axis units and added the factor from eq. 9. We removed the typo in the x-axis labels 5 (c, d) & Fig. 7 (b-g). We replaced the triangles and squares that mark the maxima in 5(d) and 7(b-g) with vertical lines and commented on it in captions.
- Figures 6, 8: We removed the typo on the x-axis labels

---

## Editorial Decision

published